# Cocoa Shell Extract Restores Redox Balance in Developmental Hypertension in Male Rats: Roles of Nrf2, SOD2 and p-eNOS

**DOI:** 10.3390/pathophysiology32040049

**Published:** 2025-09-23

**Authors:** Santiago Ruvira, Pilar Rodríguez-Rodríguez, Metee Iampanichakul, Lucía G. Cuquerella, David Ramiro-Cortijo, Silvia M. Arribas

**Affiliations:** 1Department of Physiology, Faculty of Medicine, Universidad Autónoma de Madrid, 28029 Madrid, Spain; 2Food, Oxidative Stress and Cardiovascular Health (FOSCH) Research Group, Universidad Autónoma de Madrid, 28049 Madrid, Spain; 3Department of Physiology, Faculty of Medicine, Khon Kaen University, Khon Kaen 40002, Thailand

**Keywords:** cocoa shell extract, mitochondrial SOD, supplementation, fetal programming, hypertension, oxidative damage

## Abstract

**Background and objectives**: Hypertension is a worldwide burden, for which fetal malnutrition is a risk factor. Another societal challenge is environmental waste. Our research focusses on cocoa shell extract (CSE), a cocoa by-product with antioxidant bioactive components. Male rats exposed to fetal malnutrition develop hypertension and endothelial dysfunction, which are improved by CSE supplementation. We hypothesized that effects of CSE are related to an antioxidant action. **Methods**: Adult male and female offspring of dams exposed to 50% food restriction during gestation (MUN) and controls were supplemented for 3 weeks with CSE (250 mg/kg/day) or a vehicle. We assessed plasma SOD activity, GSH and carbonyls (via spectrophotometry) and aortic expression of enzymes related to ROS degradation or production (via Western blotting). **Results**: MUN males showed lower Nrf2 expression and increased carbonyls, SOD activity and mitochondrial SOD2 expression, without alterations in GSH or the related enzyme CGLM. No changes in xanthine oxidase or NADPH subunits (p22phox and p47phox) were detected, suggesting a different origin of superoxide anion. Phosphorylated-eNOS/eNOS and 3-nitrotyrosine expression were increased without changes in plasma nitrates. MUN females only showed plasma SOD and aortic 3-nitrotyrosine elevation. CSE supplementation reduced SOD2 and p-eNOS/eNOS expression and SOD activity and increased Nrf2 expression. **Conclusions**: MUN arteries exhibit oxidative damage, with a higher impact on males. SOD2 and p-eNOS/e-NOS overexpression may be a counteracting mechanism that compensates for superoxide anion overproduction, likely involving mitochondria. The reversal of these alterations by CSE supplementation is probably related to a reduction in vascular superoxide anion through a direct scavenging action of its bioactive components. A longer supplementation period may be needed to increase endogenous antioxidants through Nrf2 and to reduce oxidative–nitrosative damage.

## 1. Introduction

Exposure to stress factors during the fetal period, such as toxic substances and nutrients or oxygen deprivation, creates an adverse intrauterine environment with a negative impact on fetal development, leading to low birth weight. Exposure to these factors also alters fetal gene expression, increasing the risk of hypertension and cardiovascular diseases (CVDs) in adult life, a process known as fetal or developmental programming. The importance of the intrauterine period for future health has been well established in human populations exposed to the abovementioned stressors during gestation [1,2] and confirmed in animal models, which have also been useful in investigating the mechanisms implicated [3]. There is ample evidence that redox signaling disruption, a well-established central mechanism in hypertension and CVDs [4], also participates in the response of the fetoplacental unit to an adverse intrauterine environment and contributes to the increased cardiovascular risk in offspring [5,6,7].

The central role of oxidative stress in CVDs has motivated the interest in the use of food-derived antioxidants as a strategy to reduce the burden of these non-communicable diseases. Cocoa (*Theobroma cacao* L.) products are of interest, since it has been demonstrated that dark chocolate consumption decreases the risk of coronary heart disease, stroke and diabetes [8,9], with these effects being related to their content of flavanols, which have antioxidant and anti-inflammatory properties [10]. The cocoa food industry generates large amounts of residues from parts of the plant which are mostly underutilized and rich in bioactive compounds. These by-products can be upcycled into ingredients and nutraceuticals targeting non-communicable diseases while contributing to sustainable development. We have previously studied the cocoa shell, a by-product generated during cocoa bean roasting, from which we obtained an aqueous extract (cocoa shell extract, CSE). This extract has been previously fully characterized as having a high content of methylxanthines (caffeine and theobromine) and phenolic compounds, such as hydroxybenzoic acids, flavan-3-ols, flavonols and flavones [11]. CSE and its main bioactive components have superoxide anion scavenging and vasodilatory actions ex vivo [12]. CSE administration has no toxic effects on rodents, even at high doses [11], and supplementation for 1 week at 250 mg/kg daily in rats substantially modifies the rats’ plasma metabolic profile, with its bioactive ingredients being detectable [13]. In our previous study, using adult rats exposed to fetal malnutrition which developed hypertension, 3-week CSE supplementation at the above-mentioned dose was shown to have blood pressure lowering effects, improving vasodilatation [14]. The present work aimed to evaluate, in the same animals, if these effects are related to an antioxidant action, either directly through its bioactive compounds or indirectly through modifying the expression of enzymes related to production or elimination of reactive oxygen species (ROS) and nitric oxide (NO). We used the rat model of fetal programming induced by maternal malnutrition in gestation since the offspring exhibit oxidative alterations in cardiovascular tissue, mostly affecting males [15].

## 2. Materials and Methods

### 2.1. Experimental Animal Model

Experiments were performed on eight- to ten-month-old male and female Sprague Dawley rats. The rats were obtained from the Animal House Facility of the Universidad Autónoma de Madrid (Spain; ES-28079-0000097). The experiments were approved by the Ethics Review Boards of Universidad Autónoma de Madrid (Ref. CEI-UAM 96-1776-A286) and the Regional Environment Committee of the Comunidad Autónoma de Madrid (Ref. PROEX 04/19). The experiments also conformed to the Guidelines for the Care and Use of Laboratory Animals [16], the EU Directive 2010/63/EU on the protection of animals and the Spanish legislation (RD 53/2013).

An experimental model of fetal programming of hypertension induced by maternal undernutrition during gestation (MUN) was used. The MUN model has been previously described in [14]. Briefly, after mating, day 1 of gestation was established by the presence of a vaginal plug in the cage. For the first 10 days, pregnant rats were fed ad libitum, and from day 11 until delivery, 50% of usual daily intake (previously calculated as 12 g/day) was given. Control group dams were fed ad libitum through gestation. All the dams were fed with a breeding diet (Euro Rodent Diet 22; 5LF5, Labdiet S.L., Madrid, Spain) containing 55% carbohydrates, 22% protein, 4.4% fat, 4.1% fiber and 5.4% mineral, being 0.26% sodium. Drinking water was also provided ad libitum.

After birth, both control and MUN groups were fed ad libitum with the same diet and litter size was adjusted to 12 individuals (6 males and 6 females whenever possible). After weaning at the age of 21 days, rats from the same sex were housed in boxes with poplar bedding in type III (24 × 19 × 45 cm; length × height × width) or type IV cages (55 × 18 × 32 cm; length × height × width), in groups of 3–5 per box according to their weight. Rats were maintained under controlled conditions (12/12 light/dark photoperiod, 22 °C, 40% relative humidity) throughout the experimental period, with Animal House Facility staff regularly evaluating their health. After weaning, all pups were fed a control diet until the end of the study.

### 2.2. CSE Supplement Preparation and Supplementation Procedure

Rat supplementation with CSE was based on voluntary ingestion of gelatine cubes containing 250 mg/kg/day of CSE or a neutral gelatine (vehicle, VEH). CSE was prepared from cocoa shell, kindly supplied by Chocolates Santocildes S.A. (León, Castilla y León, Spain), through an aqueous extraction, as previously described [17]. Gelatin cubes were prepared in water with 100% bovine gelatine (Inkafoods S.L., Barcelona, Spain) at a concentration of 140 g/L, adding sucralose (0.6 g/L; sucralin, sucralose S.L., Barcelona, Spain) and vanilla flavoring (4.8 mL/L; MyProtein, Hut.com Ltd., Manchester, UK), with or without CSE at 250 mg/kg/day, calculated according to rat weight, and transferred to molds ensuring a cube size suitable for rat handling.

At the age of 9 months, control and MUN rats were familiarized with VEH cubes with a training protocol lasting around 5 days, and then they were randomized into four groups: Control-VEH, Control-CSE, MUN-VEH and MUN-CSE. Each group comprised 6 to 7 males and 6 to 7 females from 3 to 4 different dams. The rats were given the corresponding cube for 3 consecutive weeks (5 days/week). To ensure dosage, the rats were individually placed in an empty cage with the gelatine cube until they completed the intake, and then, they were returned to the commune cage. The rats were weighted (g) and humanely killed using an incremental series of CO_2_ concentrations, followed by blood extraction by cardiac puncture and exsanguination. Subsequently, several tissues were dissected for a variety of experimental procedures in accordance with the 3Rs principles. For the present study, plasma and aorta were used.

### 2.3. Plasma Biomarkers of Oxidative Status

The blood, obtained in Eppendorf tubes containing 5% heparin, was centrifugated for 10 min (900 g at 4 °C) and the plasma was carefully aliquoted and stored at −80 °C until use. The experimental procedures to evaluate plasma biomarkers have been previously published [15,18] and are described briefly below.

**Plasma protein**: Protein content was assessed by Bradford assay. The absorbance was measured at 595 nm wavelength in a microplate reader (Synergy HT Multi-Mode, BioTek, Winooski, VT, USA) using bovine serum albumin as standard.

**Protein carbonyl levels**: Carbonyl levels were evaluated with a 2,4 dinitrophenylhydrazine (DNPH) assay adapted to a microplate reader using an extinction coefficient of 2,4-dinitrophenylhydrazine (ε = 22,000 M/cm), and absorbance read at 370 nm with the microplate reader using Guanidine-HCl as standard. Carbonyl levels are expressed as nmol/mg protein. Inter-assay coefficient variability (CV) = 36.4%.

**Glutathione (GSH) quantification**: GSH level was assessed using a fluorometric method based on the signal of o-phthalaldehyde and GSH reaction, which was measured in the microplate reader at a wavelength of 360 ± 40 nm excitation/460 ± 40 nm emission. GSH levels are expressed as µmol/mg protein. Inter-assay CV = 21.4% and intra-assay CV = 2.7%.

**Superoxide dismutase (SOD) activity**: SOD activity assay Superoxide dismutase (SOD) activity was also assessed using a kit (SOD Activity Assay kit, KB-03-011, Bioquochem, Gijón, Spain), according to the manufacturer’s instructions. Absorbance was read at 450 nm in a microplate reader and SOD activity was expressed as mU SOD/mL. Inter-assay CV = 34.8% and intra-assay CV = 5.3%.

**Plasma nitrates**: The Griess reaction was used to assess nitrate concentration, adapted to a microplate reader, as previously described [15]. The final mixture was incubated for 1 h at 37 °C, and the absorbance was read at 540 nm. Nitrates were expressed as μM. Inter-assay CV = 18.3% and intra-assay CV = 5.6%.

### 2.4. Western Blotting

The abdominal aorta was used to determine the levels of pro and antioxidant enzyme expression, using glyceraldehyde-3-phosphate dehydrogenase (GAPDH) as a loading control, as previously described [15]. Abdominal aortas were sliced into 0.2 g segments and then homogenized with lysis buffer. Following tissue homogenization, centrifugation at 10,000 rpm at 4 °C for 10 min was performed, and the supernatant was transferred into an Eppendorf tube and stored at −80 °C until use. A Bradford assay was performed to quantify the protein content of the sample as described above. Every sample was transferred into an SDS-PAGE gels and separated by electrophoresis using Mini Protean 3 (Bio-Rad; Hercules, CA, USA) in an electrophoresis buffer (0.2 M glycine; 0.025 M Tris y 0.1% SDS) and finally transferred into a polyvinylidene fluoride (PVDF) membrane. Primary antibodies were incubated overnight at 4 °C, diluted in 5% BSA-PBS-T solution. The protein concentration of the sample, loaded percentage of acrylamide gel and primary antibody concentration are represented in Table 1.

The corresponding secondary antibody (1:10,000; anti-rabbit or anti-mouse IgG, peroxidase-conjugated) was incubated for 1 h at room temperature with a 5% BSA-PBST solution. Protein expression was detected by chemiluminescence with an acquisition system (ChemiDoc XRS, Bio-Rad, USA) and previous incubation of the membranes with commercially enhanced chemiluminescence reagents (ECL Prime, Amersham Bioscence, Amersham, UK) for 5 min. Quantification of protein expression was assessed with Image Lab 3.0 (Bio-Rad, USA). Data were normalized with control groups and expressed as percentages.

### 2.5. Statistical Analysis

Statistical analysis was performed with the R software (version 3.6.0, 2018, R Core Team, Vienna, Austria) within the RStudio interface using the *rio*, *dplyr*, *compareGroups*, *ggpubr*, *devtools*, *ggplot2*, *effectsize*, *rstatix* and *pwr* packages. The Kolmogorov–Smirnov test was used to evaluate data distribution. Since several parameters did not follow a normal distribution, data are presented as boxplots with median and interquartile range [Q1; Q3].

Given the previous evidence of the influence of sex on the oxidative stress in MUN model [15], the data were analyzed subset by sex. The Mann–Whitney U test was used to evaluate the effect of CSE between groups (control–VEH versus control–CSE; MUN-VEH versus MUN-CSE). Additionally, this non-parametric test was also used to analyze the effect of MUN model on oxidative status (control–VEH versus MUN-VEH).

The statistical power was above 80% in all cases. The *p*-value (*p*) < 0.05 was considered statistically significant. For significant variables, the absolute effect size values (rank-biserial correlation similar to Cohen’s d for non-parametric test) were calculated.

## 3. Results

### 3.1. Animal Characteristics

In this study the same animals reported in [14] were used, where we demonstrated that adult MUN males were hypertensive and CSE supplementation decreased systolic blood pressure.

No significant differences in body weight were observed between groups, either in males or in females, after supplementation (Table 2).

### 3.2. Plasma Oxidative Status

In male rats (Figure 1A), MUN-VEH exhibited increased plasma carbonylated proteins compared with control–VEH (*p* = 0.005; d = |0.58|), indicative of protein oxidative damage. CSE supplementation did not modify carbonyl levels, either in MUN or control groups. No differences in plasma GSH levels were observed between MUN and control male rats and CSE supplementation did not induce significant modifications. MUN-VEH males had elevated plasma SOD activity compared to control–VEH (*p* = 0.003; d = |0.63|). CSE administration resulted in significant SOD activity reduction in MUN-CSE group (*p* = 0.014; d = |0.78|) with no effect on control male rats.

In females (Figure 1B) no significant difference in plasma carbonyl levels were observed between MUN-VEH and control–VEH rats, and CSE supplementation did not modify carbonyls in either MUN or control groups. Regarding plasma GSH, no differences were found between MUN-VEH and control–VEH females. CSE supplementation significantly reduced GSH levels in the control group (*p* = 0.004; d = |1.0|), with no effect on MUN females. Like males, MUN-VEH female plasma showed higher SOD activity than control–VEH (*p* = 0.020; d = |0.83|). However, CSE supplementation had no effect on SOD activity.

### 3.3. Antioxidant Enzyme and Nrf2 Expression in Abdominal Aorta

In males (Figure 2A), the MUN-VEH group exhibited a reduction in Nrf2 protein expression in comparison to control–VEH group (*p* = 0.017; d = |0.76|). CSE supplementation significantly increased Nrf2 in MUN-CSE males (*p* = 0.012; d = |0.93|), while it did not modify expression in control–CSE males. No differences in CGLM were detected between MUN-VEH and control–VEH groups. CSE supplementation reduced GCLM in control–CSE males (*p* = 0.057; d = |0.88|). SOD2 expression was higher in MUN-VEH males compared to control–VEH (*p* = 0.004; d = |1.0|) and CSE supplementation significantly reduced it (*p* = 0.008; d = |1.0|), without an effect on the control–CSE group.

In females (Figure 2B), no significant difference was detected between MUN-VEH and control–VEH in Nrf2, CGLM or SOD2 protein expression, and CSE supplementation did not influence the expression of any of these enzymes.

### 3.4. Expression of Enzymes Related to ROS Production in Abdominal Aorta

In males (Figure 3A), there was no significant difference in XO protein expression between MUN-VEH and control–VEH rats and CSE supplementation did not elicit an effect in any of the groups. Similar results were obtained with the two subunits of NADPH oxidase p22phox and p47phox.

In females (Figure 3B), no significant differences were found between control—VEH and MUN-VEH groups in any of the enzymes evaluated, and CSE supplementation did not elicit modifications in any group.

### 3.5. Expression of eNOS and 3-Nitrotyrosine in Abdominal Aorta and Plasma Nitrates

In males (Figure 4A), the MUN-VEH group showed significantly larger (p-eNOS/total eNOS) protein expression, which represents active enzyme, compared to the control–VEH group (*p* = 0.051; d = |0.81|). CSE supplementation increased p-eNOS/eNOS expression only in control–CSE group (*p* = 0.017; d = |0.94|), without an effect on MUN-CSE males. In MUN-VEH males, 3-NT expression was significantly larger compared to control–VEH (*p* = 0.048; d = |0.71|). CSE supplementation significantly reduced 3-NT in the control–CSE group (*p* = 0.012; d = |0.93|), without effect on MUN-CSE. Plasma nitrates were not different between MUN-VEH and control–VEH males and CSE supplementation did not modify the levels in either group.

In females (Figure 4B), no differences in p-eNOS/eNOS expression were detected between the MUN-VEH and control–VEH groups and CSE supplementation did not modify protein expression. In MUN-VEH females, 3-NT expression was significantly larger compared to control–VEH (*p* = 0.027; d = |0.62|). CSE supplementation did not modify expression in either control or MUN rats. Plasma nitrates were not different between MUN-VEH and control–VEH females and CSE supplementation did not modify the levels in either group.

## 4. Discussion

Malnutrition during gestation in experimental animals is one of the best-known stressors linked with CVD in the descendants, with a larger impact on male offspring [19]. In our previous studies using the MUN model, we have demonstrated the development of hypertension in males in adult age [15]. We have also shown that supplementation of MUN adult males with CSE reduced blood pressure through an improvement of vasodilatation [14]. The present study aimed to evaluate, in the same rats, whether these effects are mediated through modifications in oxidative status by the extract. Our main conclusion is that CSE exerts its antihypertensive and vasodilatory effects through the reduction in arterial superoxide anion, rather than through the up-regulation of antioxidant enzymes. This effect is likely related to a direct ROS scavenging action of CSE bioactive components or to an action at the mitochondrial level (Figure 5).

The present study demonstrates an elevation of oxidative and nitrosative damage in 10-month-old MUN males, like the findings in other models of maternal malnutrition [20,21] and prenatal hypoxia [22,23], confirming the central role of oxidative stress in fetal programming [24]. This is likely related to greater superoxide anion production, which we have demonstrated in resistance arteries from the same rats used in the present study [14]. We did not assess ROS in the aorta; however, since we have evidence of elevation of superoxide anion in the iliac and carotid arteries [15], we think that increased ROS production is a common feature in the vasculature of MUN males. The lower level of vascular oxidative stress in females observed in the present study, together with our previous reports of lower superoxide anion production, may contribute to the preservation of a better function and protection from hypertension development. However, 10-month-old MUN females exhibited increased 3-NT levels, not found previously in younger females [15], suggesting a deterioration of the cardiovascular system over aging. It is possible that this is, at least in part, related to a decline in estrogens, since they modulate the activity of several cardiovascular regulatory systems, and their protective role in the fetal programming of hypertension has been evidenced in several animal models of developmental insult [25,26]. Even though CSE supplementation diminished vascular superoxide anion in MUN rats, it could not reduce significantly oxidative or nitrosative damage. In Zucker diabetic rats, cocoa-based supplementation enriched in flavonoids decreased protein oxidation only after 12 weeks of administration [27]. Therefore, it is possible that 3 weeks of CSE supplementation was not sufficient to induce a significant removal of tissue damage, and a longer period may be needed.

We hypothesized that excessive superoxide anion production observed in MUN rats could be generated by NAD(P)H oxidase or XO, as both proteins are important ROS vascular sources, and they are up regulated in hypertension [28,29,30]. In addition, elevation in XO and NADPH oxidase have been found in several models of fetal programming of hypertension [21,31]. Nevertheless, in aortic tissue we did not detect any significant change in protein expression of these enzymes. An alternative source of superoxide anion may be the mitochondria, since alterations in respiratory chain due to an inefficient electron transport system have been found in fetal programming [32,33,34,35]. MUN rat aorta exhibited an up-regulation of SOD2 and Mn-SOD of mitochondrial origin. We suggest that increased SOD2 expression and plasma SOD elevation reflects a compensatory mechanism due to excess mitochondrial ROS production. This is supported by previous reports showing similar up-regulations in a model of uterine artery ligation [36], in the placenta from small for gestational age children [37] and in a mouse model of fetal malnutrition [38].

Antioxidants are currently considered as a possible strategy to reduce hypertension of developmental origins [7] and we hypothesize that CSE methylxanthines and polyphenols, with proven antioxidant actions, may contribute to the blood pressure lowering and vasodilatory effects in MUN model. We evaluated if CSE supplementation exerted an antioxidant effect as direct free radical scavenger or inducing expression of endogenous antioxidant enzymes. In arterial tissue, CSE and its main components—protocatechuic acid, caffeine and theobromine—effectively scavenged superoxide anion [12]. Similar effects have been previously demonstrated for protocatechuic acid in aortic cells from hypertensive rats [39] and caffeine on hydroxyl radicals [40]. Alternatively, CSE could increase the expression of antioxidant enzymes through Nrf2 stimulation, a transcription factor that binds to Antioxidant Response Elements, which is upregulated by tea polyphenols [41], resveratrol [42] or curcumin [43]. We found that Nrf2 was reduced in aorta from MUN males but not females, highlighting a sex-dependent response to fetal malnutrition. Even though CSE supplementation increased Nrf2, it did not up-regulate the antioxidant enzymes studied, which were unchanged or even decreased. In MUN rats, CSE supplementation reduced Mn-SOD expression and concomitantly plasma SOD levels, which were increased in MUN compared to control rats. Song et al. also demonstrated that tea polyphenols were able to increase Nrf2 protein levels while decreasing SOD2 expression [41]. Protein expression of GCLM, the rate-limiting enzyme involved in GSH synthesis, was not altered in MUN rats, as previously reported in other fetal malnutrition model [20] and CSE supplementation did not modify its expression. CSE reduced circulating GSH in control females, in line with the reduction in glutathione metabolism observed after 1-week CSE supplementation [13]. Taking together the present data and our previous reports, we suggest that the main effect of CSE supplementation is likely a reduction in ROS, which may mitigate tissular oxidative challenge, leading to a subsequent reduction in antioxidant enzymes expression. Since NADPH oxidase or XO were not modified by supplementation, we propose direct superoxide anion scavenging actions of the CSE bioactive components, which we have demonstrated to be present in rat plasma after 1 week supplementation [13]. We cannot discount an effect at the mitochondrial level, since catechins and their derivatives, also contained in CSE, exert direct modulation of mitochondrial functions [44]. The effective blood pressure reduction in these rats, despite the lack of effect of CSE on aortic oxidative and nitrosative status, supports possible additional pathways, including the modulation of the renin–angiotensin–aldosterone system, which is altered in several models of developmental programming [31,45].

In the rats used in the present study, we found an improvement of endothelium-dependent and -independent vasodilatation [14]. Since CSE had ROS scavenging actions, we proposed that it may protect NO from degradation, increasing availability. This effect has been demonstrated by protocatechuic acid in HUVEC [39] and in arterial tissue from aged rats ex vivo [12]. In addition, dietary polyphenols can stimulate the expression and/or functionality of eNOS [46]. CSE was able to increase the active phosphorylated form of the enzyme in control rats, as previously described in human coronary artery endothelial cells by epicatechin, one of the major cocoa flavonoids [47], and by methylxanthines [48]. However, this effect was not observed in MUN rats, which may be related to the fact that the enzyme was already overexpressed. We propose that this is also a compensatory mechanism due to the higher superoxide anion levels leading to NO destruction. This is evidenced by the lack of elevation of plasma NOx or arterial NO—shown in our previous study [14]—and higher aortic 3-NT, suggesting that NO is being locally eliminated by superoxide anion, leading to peroxynitrite and further protein nitrosylation.

In summary, the present data support that in MUN rats the main action of 3-week CSE supplementation may be a direct ROS scavenging action of its main bioactive components rather than stimulation on endogenous antioxidants. This is plausible, since circulating levels of methylxanthines have been found in rat plasma with only 1 week supplementation [13]. In the context of antioxidant supplementation, it must be noted that an excess of plasma antioxidants could be deleterious, since excessive polyphenols may lead to autooxidation and subsequent superoxide anion and hydrogen peroxide production, particularly if copper and iron ions are high. Other possible side effects may be the overexpression of antioxidant enzymes dramatically decreasing ROS signaling [49]. We do not think the doses used in the present study could have damaging actions since CSE supplementation at 2000 mg/kg (single dose) and sub-chronic treatment (1000 mg/kg/day for 3 months) in mice did not induce toxic effects [11].

### Limitations and Further Studies

We did not detect an improvement in endogenous antioxidants or oxidative damage with CSE, as expected, and the lack of statistical differences between groups may be related to insufficient sample size. Due to ethical constraint, it was not possible to increase the number of litters, which is a limitation. In addition, it would have been desirable to perform a direct evaluation of superoxide anion generation and to explore the respiratory chain in aorta to confirm the effects of CSE on mitochondrial ROS generation, which was not possible due to lack of adequate technology.

An aspect that requires further investigation is the effect of a longer duration of supplementation, which may induce modifications at the enzymatic level through Nrf2 pathway. In addition, although no modifications in body weight were found with CSE supplementation, based on its bioactive components, it would also be interesting to evaluate if a longer treatment could modify body composition and fat, which may exert an influence on inflammatory milieu and subsequently on the cardiovascular system.

## 5. Conclusions


CSE supplementation reverses oxidative balance alterations in MUN rats, probably related to its capacity to reduce vascular superoxide anion. This effect may contribute to previously reported antihypertensive and vasodilatory actions.CSE supplementation exerted larger effects on MUN male rats which exhibit a higher level of oxidative damage and hypertension.The up-regulation of vascular Nrf2 by CSE without effect on antioxidant enzyme protein expression suggests that a longer treatment may be needed to induce these actions.


## Figures and Tables

**Figure 1 pathophysiology-32-00049-f001:**
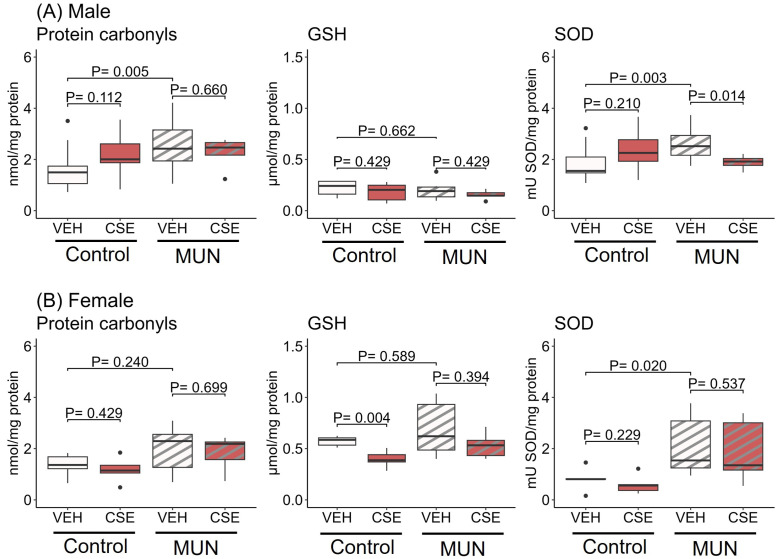
Effect of CSE supplementation on plasma carbonylated proteins, GSH and SOD activity in male (**A**) and female (**B**) MUN and control rats. Data were assessed in control and MUN adult male rats after 3-week supplementation with a neutral gelatine (vehicle, VEH) or cocoa shell extract (CSE). Data shows the median with interquartile range [Q1; Q3] and *p*-value (*p*) extracted from Mann–Whitney U test; n = six to seven rats/group from three to four dams. MUN—maternal undernutrition; GSH—reduced glutathione; SOD—superoxide dismutase. The dots show outlier values. Boxes in brown color represent rats supplemented with CSE and boxes with stripes represent MUN rats.

**Figure 2 pathophysiology-32-00049-f002:**
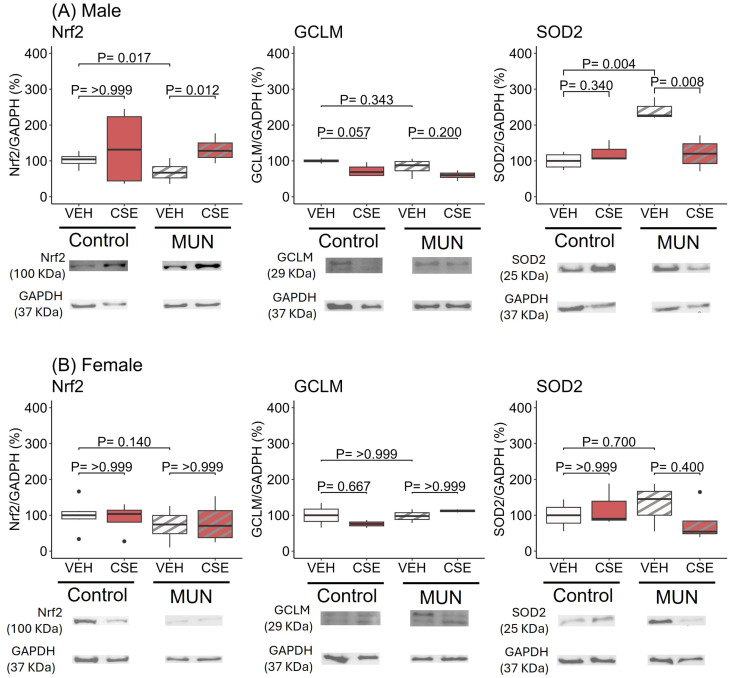
Effect of CSE supplementation on protein expression of Nrf2 and enzymes related to antioxidant production in male (**A**) and female (**B**) MUN and control rats. Data were assessed in abdominal aorta from control and MUN adult rats after 3-week supplementation with a neutral gelatine (vehicle, VEH) or cocoa shell extract (CSE). Data shows the median with interquartile range [Q1; Q3] of protein expression in relation to GAPDH content. The *p*-value (*p*) was extracted from the Mann–Whitney U test; n = four to six rats/group from three to four dams. MUN—maternal undernutrition; GCLM—glutamate-cysteine ligase modifier subunit; GAPDH—glyceraldehyde-3-phosphate dehydrogenase; Nrf2—nuclear factor erythroid 2-related factor 2; SOD 2—superoxide dismutase 2. The dots show outlier values. Boxes in brown color represent rats supplemented with CSE and boxes with stripes represent MUN rats.

**Figure 3 pathophysiology-32-00049-f003:**
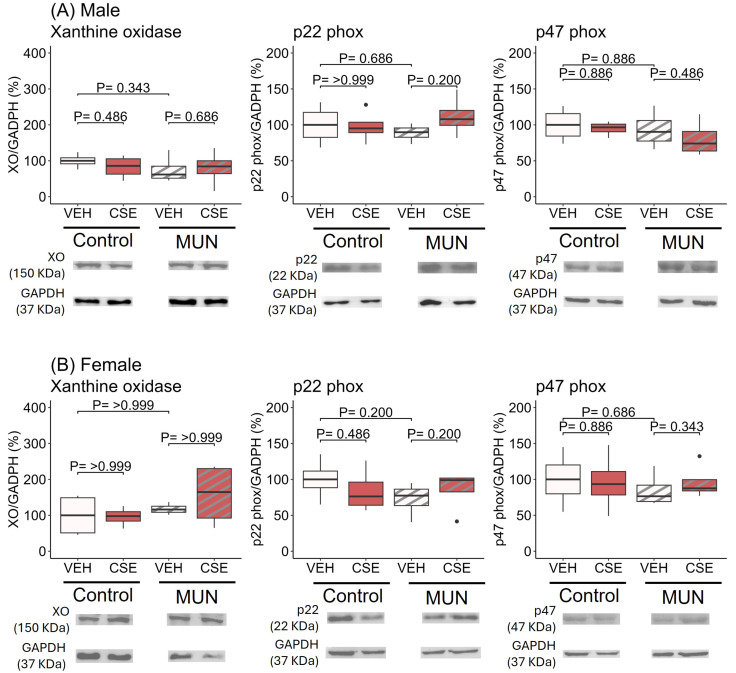
Effect of CSE supplementation on protein expression of enzymes related to ROS production in male (**A**) and female (**B**) MUN and control rats. Data were assessed in abdominal aorta from control and MUN adult rats after 3-week supplementation with a neutral gelatine (vehicle, VEH) or cocoa shell extract (CSE). Data shows the median with interquartile range [Q1; Q3] of protein expression in relation to GAPDH content. The *p*-value (*p*) was extracted from the Mann–Whitney U test; n = four to six rats/group from three to four dams. MUN—maternal undernutrition; XO—xanthine oxidase. The dots show outlier values. Boxes in brown color represent rats supplemented with CSE and boxes with stripes represent MUN rats.

**Figure 4 pathophysiology-32-00049-f004:**
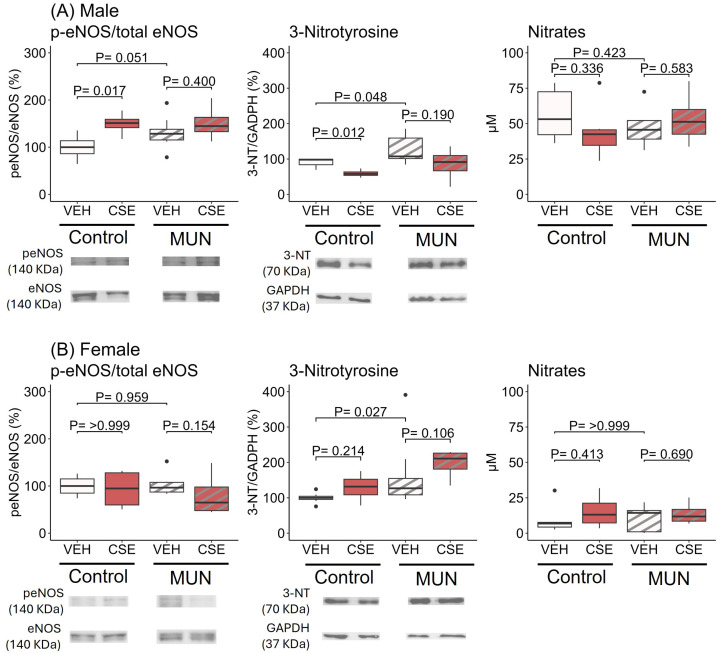
Effect of vehicle or CSE supplementation on protein expression of p-eNOS/eNOS, and 3-NT and plasma nitrates in MUN and control male (**A**) and female (**B**) rats. Data were assessed in abdominal aorta and in plasma from control and MUN adult rats after 3-week supplementation with a neutral gelatine (vehicle, VEH) or cocoa shell extract (CSE). Data shows the median with interquartile range [Q1; Q3] of protein expression in relation to GAPDH content. The *p*-value (*p*) was extracted from the Mann–Whitney U test; n = four to six rats/group from three to four dams. MUN—maternal undernutrition; eNOS—endothelial nitric oxide; p-eNOS—phosphorylated endothelial nitric oxide; 3-NT—3-nitrotyrosine. The dots show outlier values. Boxes in brown color represent rats supplemented with CSE and boxes with stripes represent MUN rats.

**Figure 5 pathophysiology-32-00049-f005:**
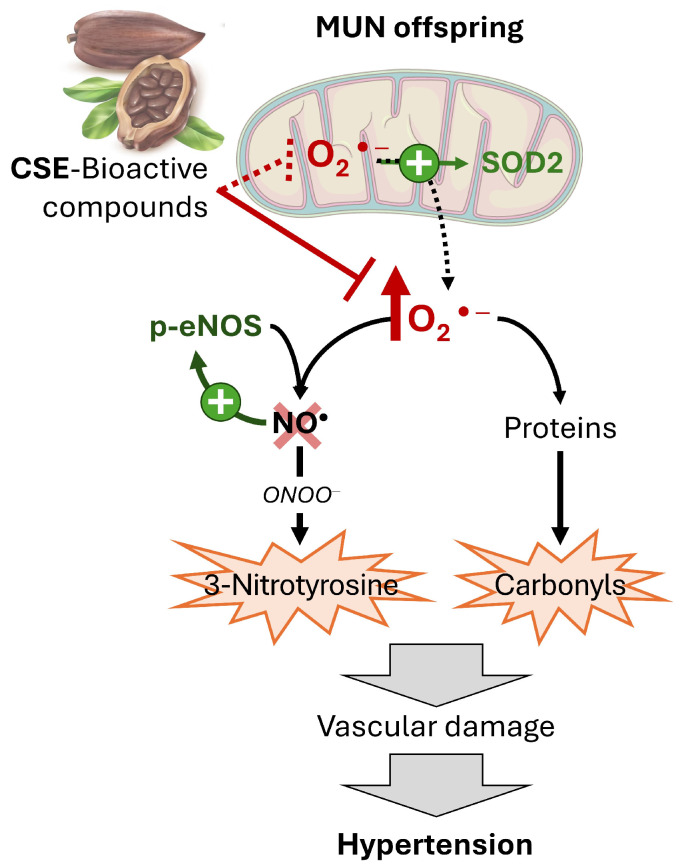
Proposed effect of CSE on oxidative status in MUN rat model of developmental hypertension. MUN offspring exhibits excess vascular superoxide anion (O_2_^•−^), which induces nitrosative and oxidative damage and reduces NO bioavailability, this reduction leads to up-regulation of p-eNOS and SOD2 as compensatory mechanisms, but insufficient to reduce vascular dysfunction. CSE bioactive compounds reduce the excess superoxide anion (O_2_^•−^) either as direct scavengers or acting at mitochondrial level (shown by dashed line). CSE-cocoa shell extract; MUN—maternal undernutrition; p-eNOS—phosphorylated endothelial nitric oxide; SOD2—superoxide dismutase type 2.

**Table 1 pathophysiology-32-00049-t001:** Antibodies and Western blot conditions.

Antibody	Loading	SDS-PAGE	Dilution	MW (KDa)	Company (Country)
Nrf2	30	12%	1:1000	90	Abcam (Cambridge, UK)
Xanthine oxidase	30	12%	1:1000	150	Santa Cruz Biotechnology (Dallas, TX, USA)
p22phox	30	12%	1:500	22	Santa Cruz Biotechnology (Heidelberg, Germany)
p47phox	30	12%	1:500	47	Santa Cruz Biotechnology (Heidelberg, Germany)
SOD2	30	12%	1:1000	25	Santa Cruz Biotechnology (Dallas, TX, USA)
GCLM	30	12%	1:500	29	Invitrogen (Carlsbad, CA, USA)
eNOS	30	7%	1:250	140	BD Transduction (San Jose, CA, USA)
p-eNOS (Ser1177)	30	7%	1:250	140	Cell Signalling Technology (Danvers, TX, USA)
3-Nitrotyrosine	25	8%	1:1000	50	Abcam (Waltham, MA, USA)
GAPDH	30	12%	1:5000	37	Cell Signalling Technology (Danvers, TX, USA)

MW—molecular weight; Nrf2—nuclear factor (erythroid-derived 2)-like 2; NADPH—oxidase subunits p22phox and p47phox; SOD2—superoxide dismutase-2 (Mn-SOD); GCLM—cysteine–glutamate ligase modifier subunit. eNOS—endothelial nitric oxide synthase; p-eNOS—phosphorylated endothelial nitric oxide synthase; GAPDH—glyceraldehyde 3-phosphate dehydrogenase.

**Table 2 pathophysiology-32-00049-t002:** Body weight after supplementation in male and female MUN and control rats.

Weight (g)	Control		MUN		*p* ^1^	*p* ^2^	*p* ^3^
	VEH	CSE	VEH	CSE			
Male	478 [441; 495]	498 [485; 502]	525 [476; 562]	456 [444; 469]	0.682	0.067	0.125
Female	303 [284; 304]	289 [287; 302]	304 [297; 316]	284 [281; 288]	0.991	0.250	0.975

Data were assessed in control and MUN adult male rats after 3-week supplementation with a neutral gelatine (vehicle, VEH) or cocoa shell extract (CSE). Data shows the median with interquartile range [Q1; Q3] and *p*-value (*p*) was extracted from the Mann–Whitney U test. *p* ^1^: Control–VEH vs. control–CSE; *p* ^2^: MUN-VEH vs. MUN-CSE; *p* ^3^: control–VEH vs. MUN-VEH; n = six to seven rats/group from four dams. MUN—maternal undernutrition.

## Data Availability

The original contributions presented in the study are included in the article; further inquiries can be directed to the corresponding author.

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
