# Peer review of "Cocoa Shell Extract Restores Redox Balance in Developmental Hypertension in Male Rats: Roles of Nrf2, SOD2 and p-eNOS"

_pathophysiology, 2025, doi:10.3390/pathophysiology32040049_

Round 1

Reviewer 1 Report

Comments and Suggestions for Authors

This manuscript studies the maternal undernutrition (MUN), inducing fetal programming of cardiovascular disease (CVD) by altering redox signaling and gene expression, leading to hypertension and oxidative stress in adulthood. The authors have shown previously that Cocoa shell extract (CSE), rich in flavanols and methylxanthines, improves antioxidant capacity and decrease blood pressure in a rat MUN model. This study further investigates whether CSE supplementation (250 mg/kg/day for 3 weeks) benefits are related to direct or indirect antioxidants effects.

MUN offspring exhibited elevated systolic blood pressure (SBP), which CSE supplementation significantly reduced. No effect was observed in females.

MUN males showed early antioxidant deficits and oxidative damage in cardiovascular tissues. CSE modulated plasma biomarkers (protein carbonyls, glutathione, SOD activity) and aortic expression of redox-related enzymes (e.g., NOX4, SODs).

Authors conclude that CSE’s antihypertensive effects may involve direct ROS scavenging and regulation of antioxidant enzyme expression; and that CSE ameliorates developmental programming of hypertension via redox modulation, supporting its potential as a nutraceutical for CVD prevention.

Major comments:

  1. When designing an experiment involving animal offspring, the sample size (n) typically refers to the number of independent experimental units—which, in this case, are usually the mothers (dams), not the individual babies (pups). It seems that some of experiments include siblings in the same experimental group as the authors declare 3-4 dams per group and 6-7 offspring individuals, then the true biological “n” is 3-4.
  2. Have authors estimate the sample size (n) needed to achieve statistical significance by a power analysis? There are few differences in-between groups and this might represent not enough chance (power) to detect a true effect if it exists.
  3. Authors conclude that CSE induce a decreased Superoxide anion generation; however, they did not measure the production, not even de amount in the tissue. Do they have any chances to measure either O2- generation, protein activity associated to O2- generation (ie. NADPH, mitochondrial respiratory chain, etc.), antioxidant capacity of CSE and/or antioxidant enzymatic activity. Incorporating these or similar methods may give results to support their conclusions.

Minor observations:

Typos. line 91: delete “was used”;   line 554: “whith”

Author Response

This manuscript studies the maternal undernutrition (MUN), inducing fetal programming of cardiovascular disease (CVD) by altering redox signaling and gene expression, leading to hypertension and oxidative stress in adulthood. The authors have shown previously that Cocoa shell extract (CSE), rich in flavanols and methylxanthines, improves antioxidant capacity and decrease blood pressure in a rat MUN model. This study further investigates whether CSE supplementation (250 mg/kg/day for 3 weeks) benefits are related to direct or indirect antioxidants effects.

MUN offspring exhibited elevated systolic blood pressure (SBP), which CSE supplementation significantly reduced. No effect was observed in females.

MUN males showed early antioxidant deficits and oxidative damage in cardiovascular tissues. CSE modulated plasma biomarkers (protein carbonyls, glutathione, SOD activity) and aortic expression of redox-related enzymes (e.g., NOX4, SODs).

Authors conclude that CSE’s antihypertensive effects may involve direct ROS scavenging and regulation of antioxidant enzyme expression; and that CSE ameliorates developmental programming of hypertension via redox modulation, supporting its potential as a nutraceutical for CVD prevention.

Response: Thank you for taking the time to review our manuscript and for your helpful comments. Below you can find our point-by-point responses and edits within the main text of the manuscript.

Major comments:

  1. When designing an experiment involving animal offspring, the sample size (n) typically refers to the number of independent experimental units—which, in this case, are usually the mothers (dams), not the individual babies (pups). It seems that some of experiments include siblings in the same experimental group as the authors declare 3-4 dams per group and 6-7 offspring individuals, then the true biological “n” is 3-4.

Response: We agree on the fact that sample size should refer to the number of dams. For clarification the figure legends now show the number of dams in addition to the number of pups.

  1. Have authors estimate the sample size (n) needed to achieve statistical significance by a power analysis? There are few differences in-between groups, and this might represent not enough chance (power) to detect a true effect if it exists.

Response: We did not calculate sample size, since we used the tissues from the same animals of our previous study (DOI: 10.1113/JP287097) for ethical reasons to maximize the use of animals (3Rs). We agree that the lack of statistical significance in some of the parameters may be related to low sample size. Since it is not possible to perform additional experiments, we have explicitly mentioned this aspect as a limitation (lines 538-540). On the other hand, regarding data showing statistical significance, we think we had sufficient power to detect differences, since for many parameters the power was above 80% and size effect >|0.8|. We have now included both values in results section.

  1. Authors conclude that CSE induces a decreased Superoxide anion generation; however, they did not measure the production, not even the amount in the tissue. Do they have any chances to measure either O2- generation, protein activity associated to O2- generation (i.e. NADPH, mitochondrial respiratory chain, etc.), antioxidant capacity of CSE and/or antioxidant enzymatic activity. Incorporating these or similar methods may give results to support their conclusions.

Response: We agree that it would be valuable to confirm our suggestion. Unfortunately, we do not have sufficient aortic tissue to perform such experiments. However, we believe that this suggestion has robust support. Firstly, we have demonstrated the in vitro superoxide anion scavenging capacity of CSE and its main bioactive components in both conduit and resistance arteries (DOI: 10.3390/antiox11020429). Secondly, we have evidence that CSE supplementation induced a reduction of superoxide anion generation in mesenteric resistance arteries (DOI: 10.1113/JP287097). Therefore, similar actions can be expected in aorta. It would have been desirable to explore the respiratory chain in aorta to confirm the effects of CSE on the mitochondria, which was not possible due to lack of adequate technology. We are aware of this limitation and have included this aspect in discussion section (lines 541-544).

Minor observations: Typos. line 91: delete “was used”; line 554: “whith”

Response: thank you for the observation. We have corrected both typos.

Reviewer 2 Report

Comments and Suggestions for Authors

Reviewer Comments

Santiago Ruvira et al. investigate the antioxidant effects of cocoa shell extract (CSE) in a rat model of fetal undernutrition. Their study demonstrates that male offspring exhibit increased oxidative stress, which is attenuated by CSE supplementation through modulation of enzymes such as SOD2, phosphorylated eNOS, and Nrf2. These findings support the potential of CSE to reduce vascular oxidative damage via its bioactive components.

Major Comments

  1. Title
    While it is acceptable to reuse animals and refer to earlier data, the title should reflect what is actually measured in the present study (i.e., oxidative stress markers and enzyme expression). Importantly, the manuscript does not include any new blood pressure data, yet references “hypertensive rats” in the title, which is misleading. The title should also clearly specify the experimental model (rats exposed to fetal undernutrition), without implying unreported results.

Suggested title:
“Cocoa Shell Extract Reduces SOD2 and p-eNOS Overexpression in Male Rats with Prenatal Undernutrition”

  1. Section 2.3 (Lines 128–137): Systolic Blood Pressure
    This section heavily relies on previously published data (Ruvira et al. [14]) and does not present any new results on blood pressure. Including it without new analysis is redundant. I recommend removing this section entirely. A brief reference to prior findings can instead be included in the introduction or discussion to provide context for exploring molecular mechanisms.
  2. Line 117 – Dose Administration
    The manuscript states that CSE was administered at 250 mg/kg/day via supplemented cubes. However, it is unclear how consistent dosing was ensured, given that animals were group-housed and had free access to the cubes. Please clarify how individual intake was controlled or estimated to ensure accurate dosing per rat.
  3. Statistical Analysis
    The current analysis is based solely on nonparametric tests (Mann–Whitney U), despite the experimental design involving two independent factors (e.g., sex and treatment). A two-way ANOVA (or its nonparametric equivalent, such as aligned ranks transformation ANOVA) would be more appropriate to test for main effects and interactions. The rationale for choosing the current approach should be provided, and reanalysis may be warranted to account for factorial interactions.

Minor Comments

  1. Line 91–92 (Grammar):
    The phrase "was used" is unnecessarily repeated.
    Suggested revision:
    "An experimental model of fetal programming of hypertension induced by maternal undernutrition during gestation (MUN) was used."
  2. Line 151 – Subsection Heading:
    "Reduced Glutathione (GSH)" should be revised to "Glutathione (GSH) Quantification" to better reflect the method described.
  3. Error Bars in Figures:
    The figure legends and methods should clarify whether standard deviation (SD) or standard error (SE) was used in graphical error bars. This is important for accurate interpretation of variability.

Author Response

Santiago Ruvira et al. investigate the antioxidant effects of cocoa shell extract (CSE) in a rat model of fetal undernutrition. Their study demonstrates that male offspring exhibit increased oxidative stress, which is attenuated by CSE supplementation through modulation of enzymes such as SOD2, phosphorylated eNOS, and Nrf2. These findings support the potential of CSE to reduce vascular oxidative damage via its bioactive components.

Response: Thank you for taking the time to review our manuscript. Below you will find our point-by-point responses and edits within the main text.

Major Comments

  1. Title. While it is acceptable to reuse animals and refer to earlier data, the title should reflect what is actually measured in the present study (i.e., oxidative stress markers and enzyme expression). Importantly, the manuscript does not include any new blood pressure data, yet references “hypertensive rats” in the title, which is misleading. The title should also clearly specify the experimental model (rats exposed to fetal undernutrition), without implying unreported results.

Suggested title: “Cocoa Shell Extract Reduces SOD2 and p-eNOS Overexpression in Male Rats with Prenatal Undernutrition”.

Response: Thank you for your suggestion. We agree with the change since we have not measured blood pressure and the title you suggest is more informative and accurate. We have changed the title to: “Cocoa Shell Extract Supplementation Reduces SOD2 and p-eNOS Overexpression in Male Rats exposed to Prenatal Undernutrition”.

  1. Section 2.3 (Lines 128–137): Systolic Blood Pressure. This section heavily relies on previously published data (Ruvira et al. [14]) and does not present any new results on blood pressure. Including it without new analysis is redundant. I recommend removing this section entirely. A brief reference to prior findings can instead be included in the introduction or discussion to provide context for exploring molecular mechanisms.

Response: We agree with your suggestion. We used the tissues from the same rats as in our previous study and did not include any new blood pressure measurement. Therefore, there is no need for the section on blood pressure, which has been removed. We have included appropriate clarifications in the introduction (line 71), at the beginning of results (lines 198-200) and discussion (line 407).

  1. Line 117 – Dose Administration. The manuscript states that CSE was administered at 250 mg/kg/day via supplemented cubes. However, it is unclear how consistent dosing was ensured, given that animals were group-housed and had free access to the cubes. Please clarify how individual intake was controlled or estimated to ensure accurate dosing per rat.

Response: Thank you for your suggestion. This information was omitted, and we agree it is very important. The dose (250 mg/kg/day) was individually calculated based on rat weight. Dosage was ensured by placing the animals individually in an empty cage until they completed the gelatin cube intake, according to our developed protocol (DOI: 10.3390/ani13111827). This procedure was also applied to vehicle gelatin cubes.  This information is now included in methods section (lines 122-124).

  1. Statistical Analysis. The current analysis is based solely on nonparametric tests (Mann–Whitney U), despite the experimental design involving two independent factors (e.g., sex and treatment). A two-way ANOVA (or its nonparametric equivalent, such as aligned ranks transformation ANOVA) would be more appropriate to test for main effects and interactions. The rationale for choosing the current approach should be provided, and reanalysis may be warranted to account for factorial interactions.

Response: The rationale to use our statistical analysis approach is as follows. We have previous evidence of oxidative stress in cardiovascular tissues from MUN model of hypertension, with a sexual dimorphism (doi:10.1371/journal.pone.0171544; doi:10.1007/s13105-023-00949-1). Based on the evidence of difference between males and females, we have subset the variables by sex to avoid this factor masking the effect of CSE, which was the main objective of our study. For this reason, we evaluated the differences between groups with and without CSE supplementation separated by sex with Mann-Whiteny U test which is a non-parametric robust test. Additionally, we corroborated the observed alterations in oxidative status in MUN model by analyzing Control-VEH versus MUN-VEH.

We have now clarified this in the statistical analysis section (lines 188-195).

Minor Comments

  1. Line 91–92 (Grammar): The phrase "was used" is unnecessarily repeated.

Suggested revision: "An experimental model of fetal programming of hypertension induced by maternal undernutrition during gestation (MUN) was used."

Response: Thank you for the observation. We have eliminated the redundant “was used”

  1. Line 151 – Subsection Heading: "Reduced Glutathione (GSH)" should be revised to "Glutathione (GSH) Quantification"to better reflect the method described.

Response: This heading has been modified

  1. Error Bars in Figures: The figure legends and methods should clarify whether standard deviation (SD) or standard error (SE) was used in graphical error bars. This is important for accurate interpretation of variability.

Response: Since we used non-parametric tests, the interquartile range [Q1; Q3] was chosen as the most adequate to represent variability. This is indicated in the figure captions and in the statistical analysis section.

Reviewer 3 Report

Comments and Suggestions for Authors

This manuscript examines the mechanisms by which the intake of cocoa shell extract improves hypertension and endothelial dysfunction in male rats caused by nutritional deficiency in the fetal period. The results corroborate those reported by the authors in previous papers. However, neither this manuscript nor the previous papers disclosed data on food intake and weight change, which should be included in this manuscript. If significant differences are observed in these data between the experimental and control groups, nutritional considerations should be taken into account when interpreting the results. Additionally, the authors should pay sufficient attention to similarities with previously reported papers when revising the manuscript.

Author Response

This manuscript examines the mechanisms by which the intake of cocoa shell extract improves hypertension and endothelial dysfunction in male rats caused by nutritional deficiency in the fetal period. The results corroborate those reported by the authors in previous papers. However, neither this manuscript nor the previous papers disclosed data on food intake and weight change, which should be included in this manuscript. If significant differences are observed in these data between the experimental and control groups, nutritional considerations should be considered when interpreting the results. Additionally, the authors should pay sufficient attention to similarities with previously reported papers when revising the manuscript.

Response: Thank you for taking the time to review our manuscript. We agree on the importance of changes in body weight, particularly in body fat, which could affect the inflammatory milieu, exerting an influence on the cardiovascular system oxidative status. We did not evaluate food intake, but assessed weight, and these data are now provided in results section in Table 2 (lines 201-208). No statistical differences in weight were detected between VEH and CSE supplemented animals and therefore, we do not think it had an influence on our results. However, we cannot discard that a longer treatment may modify body composition, and we have included this aspect as future direction (lines 546-549).

Round 2

Reviewer 1 Report

Comments and Suggestions for Authors

The authors have addressed most of my comments except for the conclusions. In my opinion, the first conclusion "CSE exerts its antihypertensive and vasodilatory effects through reduction of arterial superoxide anion, likely related to a direct ROS scavenger effect of its bioactive components. We cannot discard an action at mitochondrial level." is not supported by the experimental design and the results, as they did not and will not determine superoxide anion level (directly or indirectly). Therefore, this conclusion is just especulation.

 I suggest to change this conclusion by some sentence similar to the abstract conclusion: "The reversal of these alterations by CSE supplementation is probably related to its capacity to reduce vascular superoxide anion.."

Author Response

The authors have addressed most of my comments except for the conclusions. In my opinion, the first conclusion "CSE exerts its antihypertensive and vasodilatory effects through reduction of arterial superoxide anion, likely related to a direct ROS scavenger effect of its bioactive components. We cannot discard an action at mitochondrial level." is not supported by the experimental design and the results, as they did not and will not determine superoxide anion level (directly or indirectly). Therefore, this conclusion is just speculation.

I suggest changing this conclusion by some sentence similar to the abstract conclusion: "The reversal of these alterations by CSE supplementation is probably related to its capacity to reduce vascular superoxide anion."

Response: Thank you for your consideration. We have changed the conclusion according to your suggestion to make it less speculative and focused on oxidative balance. We also mention the antihypertensive and vasodilatory actions as previously reported (lines 552-554)

“CSE supplementation reverses oxidative balance alterations in MUN rats, probably related to its capacity to reduce vascular superoxide anion. This effect may contribute to previously reported antihypertensive and vasodilatory actions”.

Reviewer 2 Report

Comments and Suggestions for Authors

I have carefully reviewed the authors’ detailed responses and the revised version of the manuscript. The authors have appropriately addressed all major and minor concerns raised in my initial review:

  • The title has been corrected to accurately reflect the measured outcomes and experimental model, avoiding misleading reference to blood pressure.
  • The redundant section on systolic blood pressure has been removed, with clarifications provided in the introduction, results, and discussion.
  • The methodology for CSE dosing has been clarified with sufficient detail to ensure reproducibility.
  • The rationale for the chosen statistical approach (Mann–Whitney U test) has been explained, with reference to prior evidence of sex differences in this model. This clarification is acceptable.
  • Minor issues, including grammar, subsection heading, and error bar representation, have all been addressed satisfactorily.

Overall, the authors have made the necessary revisions, and the manuscript has been substantially improved. I believe it is now suitable for publication in its present form.

Author Response

I have carefully reviewed the authors’ detailed responses and the revised version of the manuscript. The authors have appropriately addressed all major and minor concerns raised in my initial review:

  • The title has been corrected to accurately reflect the measured outcomes and experimental model, avoiding misleading reference to blood pressure.
  • The redundant section on systolic blood pressure has been removed, with clarifications provided in the introduction, results, and discussion.
  • The methodology for CSE dosing has been clarified with sufficient detail to ensure reproducibility.
  • The rationale for the chosen statistical approach (Mann–Whitney U test) has been explained, with reference to prior evidence of sex differences in this model. This clarification is acceptable.
  • Minor issues, including grammar, subsection heading, and error bar representation, have all been addressed satisfactorily.

Overall, the authors have made the necessary revisions, and the manuscript has been substantially improved. I believe it is now suitable for publication in its present form.

Response: Thank you for your comments which helped to improve the manuscript. 

Reviewer 3 Report

Comments and Suggestions for Authors

The academic issues pointed out have been resolved in the revised manuscript.

Author Response

The academic issues pointed out have been resolved in the revised manuscript.

Response: Thank you for your comments which helped to improve the manuscript.